# Regenerative Endodontic Procedures: An Umbrella Review

**DOI:** 10.3390/ijerph18020754

**Published:** 2021-01-17

**Authors:** Luísa Bandeira Lopes, João Albernaz Neves, João Botelho, Vanessa Machado, José João Mendes

**Affiliations:** 1Dental Pediatrics Department, Egas Moniz—Cooperativa de Ensino Superior, 2829-511 Almada, Portugal; 2Clinical Research Unit (CRU), Centro de Investigação Interdisciplinar Egas Moniz (CiiEM), Egas Moniz—Cooperativa de Ensino Superior, 2829-511 Almada, Portugal; jalbernazneves@gmail.com (J.A.N.); jbotelho@egasmoniz.edu.pt (J.B.); vmachado@egasmoniz.edu.pt (V.M.); jmendes@egasmoniz.edu.pt (J.J.M.); 3Endodontics Department, Egas Moniz—Cooperativa de Ensino Superior, 2829-511 Almada, Portugal; 4Evidenced-Based Hub, CiiEM, Egas Moniz—Cooperativa de Ensino Superior, 2829-511 Almada, Portugal

**Keywords:** regenerative, endodontics, pediatric dentistry, oral health, dental medicine, systematic review, umbrella review

## Abstract

The Regenerative Endodontic Procedure (REP) is a biologically based method in which a damaged pulp–dentin complex is replaced by a new vital tissue. This umbrella review aimed to critically assess the available systematic reviews (SRs) on REP. An electronic database search was conducted (PubMed-Medline, CENTRAL, Scielo, Web of Science, and LILACS) until December 2020. Studies were included if they were an SR on REP. The Risk of Bias (RoB) of SRs was analyzed using the Measurement Tool to Assess SRs criteria 2 (AMSTAR2). The primary outcome was the methodological quality in each specific section of REP protocols and outcomes. From 403 entries, 29 SRs were included. Regarding the methodological quality, ten studies were of critically low, three of low, fourteen of moderate, and two were rated as high quality. The quality of evidence produced by the available SRs was not favorable. Future high standard SRs and well-designed clinical trials are warranted to better elucidate the clinical protocols and outcomes of REP.

## 1. Introduction

Endodontic management of immature permanent teeth with necrotic pulp is a challenging clinical procedure. In the last few decades, these teeth have been treated by apexification procedures with the disadvantages of compromised root development, thickening of the radicular dentin, and compromised crown-to-root ratio [1]. In light of these considerations, Regenerative Endodontic Procedure (REP) is described as “biologically based procedures designed to replace damaged structure” and aims to deliver a suitable environment to promote natural regeneration/repair with a functional pulp–dentin wall and apical closure [2,3]. Therefore, REPs have the potential to increase root length, to thicken the rootwall, and to achieve apical closure [4,5,6,7,8,9,10,11,12,13]. 

REP was first established by Nygaard-Ostby in the 1960s, though with low success [6,8,14,15]. Thus, REP represents one of the most challenging and cutting-edge topics in regenerative dentistry. The European Society of Endodontology (ESE) [16] and the American Association for Endodontists (AAE) [17,18] have recently delivered position statements and clinical considerations regarding REP. Nevertheless, according to recent systematic reviews (SRs) related to REP, there is a lack of standardization of the treatment protocol between studies. 

Overall, REP is a common category of biologically based endodontic therapy known as revascularization or revitalization. REP is a treatment option that depends on the stem cells and growth factors by stimulating them to root elongation, maturation and complete apex closure, protocol applied, being the outcomes disputable in regard to the regenerated tissue. 

Therefore, the present umbrella review aimed to critically appraise the available SRs on REPs, with a particular two-fold focus: (1) quality of evidence and (2) clinical outcomes. 

## 2. Materials and Methods

The protocol for this umbrella review was defined a priori by all authors and was performed following the Preferred Reporting Items for SRs and Meta-Analyses (PRISMA) guidelines [19] (Appendix A), expanded with the guideline for SRs [20]. 

### 2.1. Study Selection

For this umbrella review, five electronic databases (PubMed-Medline, Cochrane Database of SRs, Scielo, Web of Science, and LILACS) were searched from the earliest data available until December 2020. We merged keywords and subject headings in accordance with the thesaurus of each database: (endodontic OR pulp OR tooth) AND (revitalization OR revascularization OR regenerative OR pulpal regeneration) AND (systematic review OR meta-analysis). Grey literature was searched through the OpenGrey portal (http://www.opengrey.eu). Additional relevant literature was included after a manual search of the reference lists of the final included articles. 

The electronic database search was carried out by two authors (L.L. and J.N.) independently, and the final decision for inclusion was made according to the following criteria: (1) SRs with or without meta-analysis; (2) conducted on human and animal teeth; (3) assessing clinical characteristics of REP, revitalization, revascularization, or regeneration. There were no restrictions regarding the year of publication year nor language. 

### 2.2. Information Sources Search

A predefined table was used to extract the necessary data from each eligible SR, including the first author’s name, publication year, databases searched, number of studies included, type and number of studies included (control and interventional group if applicable), interventions, tool used to assess the quality of studies, main results, and main conclusion. From each eligible SR, two researchers (Luísa Lopes and João Albernaz Neves) independently extracted the information and all disagreements were resolved through discussion with a third reviewer (Vanessa Machado). Outcomes were classified as: protocols and materials; survival outcomes; and stem cells. 

### 2.3. Risk of Bias (RoB) Assessment

RoB of the included SRs was independently assessed by two calibrated authors (L.L. and J.N.) using the Meaurement Tool to Assess SRs (AMSTAR 2) [21]. According to this tool, SRs are categorized as: High (“Zero or one non-critical weakness”); Moderate (“More than one non-critical weakness”); Low (“One critical flaw with or without non-critical weaknesses”); and Critically Low (“More than one critical flaw with or without non-critical weaknesses”). The final quality rate was obtained via the online tool (https://amstar.ca/Amstar_Checklist.php) for each study.

## 3. Results

### 3.1. Study Selection

The electronic search strategy yielded a total of 403 entries, with 174 duplicates being excluded (Figure 1). After title and abstract assessment, 35 potentially eligible full-text articles were screened. As a result, six studies were excluded with various reasons, resulting in 29 SRs that fulfilled the eligibility criteria and were included for qualitative synthesis. Further information regarding reasons for SR exclusion is available in the Appendix A, Appendix A. Inter-examiner reliability at the full-text screening was recorded as excellent (kappa score = 1.00).

### 3.2. SR Characteristics

Overall, twenty SRs without meta-analysis [4,5,6,7,8,9,12,13,22,23,24,25,26,27,28,29,30,31,32,33] and nine SRs with meta-analysis were included [10,11,14,15,34,35,36,37,38] (Table 1). Multiple sub-topics were investigated, such as REP protocols and outcomes [4,5,7,9,11,12,13,14,15,22,27,29,36], solely REP protocols [30,37], solely outcomes [6,8,10,23,28,38], and stem cells on REP [24,25,26,31,32,33,34,35]. 

The methodological characteristics are detailed in Table 1. Six SRs failed to report a defined timeframe [6,9,14,26,28,31]. Seven SRs did not report language restrictions [11,22,23,31,33,35,38], twelve applied a language restriction as an inclusion criterion [8,10,12,13,14,15,24,26,28,29,32,36,37], and the remaining had no language restriction [4,5,6,7,9,25,27,30,34].

### 3.3. RoB

Excellent inter-examiner reliability at the RoB screening was recorded (kappa score = 0.91; 95% confidence interval: 0.89–0.92). None of the included SRs fully satisfied the AMSTAR2 Criteria (Table 2). Overall, two were rated as “high quality” [15,37], fourteen as “moderate quality” [5,7,9,10,11,13,25,27,28,29,31,34,35,36], three as “low quality” [6,8,12], and ten were assessed as “critically low quality” [4,14,22,23,24,26,30,32,33,38]. Major concerns regarding methodological quality were found on the: (a) lack of information regarding the type of included studies; (b) the literature search strategy; (c) the absence of a list of excluded studies with justification; and (d) declaration of funding sources.

### 3.4. Synthesis of Results

#### 3.4.1. Clinical Protocols

Fifteen SRs investigated the efficacy of REP protocols [4,5,7,9,11,12,13,14,15,22,27,29,30,36,37]. Twelve SRs investigated human teeth exclusively [4,5,7,9,11,13,14,15,27,30,36,37] and three included studies with human and animal teeth simultaneously [12,22,29]. Overall, REP protocols reported generic removal of necrotic pulp through minimal or no mechanical instrumentation [5,7,15,22,27,28,29]. 

Nevertheless, there is no unanimity on disinfection/irrigation, intracanal medication, and scaffolds. For this reason, ESE [16] and AAE [17,18] have recently delivered position statements and clinical considerations based on the best, yet limited, available evidence.

##### Endodontic Irrigation in REP

Solely regarding the subject of endodontic irrigation, the level of evidence of the SRs is mainly of moderate quality (Table 2). However, there are no meta-analytical estimates to assess and compare different irrigation protocols and the success of REP treatment. 

In all, SRs reported several irrigating solutions at distinct concentrations and during different periods of time except two SRs [13,30]. Rossi Fedele et al. [29] suggested that sodium hypochlorite (NaOCl) at different concentrations is the irrigating solution most used in these procedures. Additionally, in the successful reported single-visit cases, the irrigation included both 2.5% NAOCl saline and 17.0% ethylenediaminetetraacetic acid (EDTA) [29]. Nevertheless, irrigation with EDTA was not commonly used in studies including animal teeth [22,29]. Concerning two-visit REP, NaOCl at 2.5% is the most employed despite disinfection using a percentage of 5.25% to 6.0% of NaOCL associated with 0.2% to 2.0% of chlorhexidine (CHX) also being established in non-vital immature permanent teeth [4,7,14]. Other irrigation methods used NaOCl in concentrations that varied from 0.5% to 8.0%, isolated or included with sterile saline, 5.0% to 17.0% EDTA, and 0.12% to 2.0% CHX [5,9,12,15,36,37]. Furthermore, Torabinejad et al. [11] and Lobato et al. [27] reported studies with irrigation protocols with 1.0% to 5.25% of NaOCl isolated or in combination with a 0.12% CHX, 0.9% saline, or 2.0% CHX isolated that are also used in REP [11,27]. Additionally, Lobato et al. [27] record their respective irrigating solution as NaOCL at 2.5% alone or in combination with 0.12% CHX and 5.0% EDTA. 

##### Intracanal Medication

In terms of intracanal medication, the level of evidence of the SRs is mostly of moderate quality as well (Table 2). Nevertheless, none were able to synthesize meta-analytical estimates comparing the usage or not of intracanal medication.

In SRs reporting the use of intracanal medication, the most frequently used was triple antibiotic paste (TAP), which consists of metronidazole, ciprofloxacin, and minocycline [9,12,13,14,15,22,27,30,36,37]. Furthermore, a modified TAP was reported, as its third element (minocycline) could be replaced with other agents (cefaclor, doxycycline, amoxicillin, or clindamycin) [4,12,15,36]. Additionally, double antibiotic paste (DAP) was reported with a variety of cocktails [12,15,30,37]. However, TAP was also reported to be conjugated with CHX and calcium hydroxide [11,15,30] or formocresol [12] as an intracanal medication.

On the other hand, other agents were assessed in a systematic way; for instance, CHX or CHX in association with iodoform [11], calcium hydroxide isolated or associated with 2.0% of CHX [4,7,9,13,15,22,30,36], formocresol [12,14,22], or even, Augmentin [14]. 

One SR defined as exclusion criterion the use of intracanal medication and therefore, such procedures were not appraised [7]. Others have included both REP with intracanal medication and without [7,11,14]. Lastly, one SR did not report intracanal medication, as the purpose was to assess single-visit REPs [29].

##### Scaffolds

Concerning the use of scaffolds, the level of evidence of the SRs is also mostly of moderate quality (Table 2). All described scaffolds were clinically successful but so far, there has been no meta-analytical evidence regarding a supplement with better results for cell-based pulp/dentin regeneration.

The analysis of scaffolds was assessed in SRs on pulp revascularization procedures of immature necrotic teeth. Seemingly, blood clot appears to be the most reported scaffold during REP [7,11,12,13,15,22,27,36,37].

Other types of scaffolds were applied in a systematic way, namely blood clot and platelet-rich plasma (PRP) with and without collagen sponge [12,27], PRP with and without collagen [11,13,15,27,36,37], PRP and beta-tricalcium phosphate with and without hydroxyapatite [5,12], PRP with hydroxyapatite [12], PRP and bone [5], polyglycolid-polylactid (PLGA) [5], platelet-rich fibrin (PRF) [15,36,37], blood clot with both PRP and PRF [12], PRF with and without blood clot [11], blood clot with collagen sponge or gelatin hydrogel [11,15], and collagen calcium phosphate gel [11], platelet pellet [15,37], polymer fleece [5], bovine bone mineral [5], and empty scaffold [12]. 

Furthermore, four SRs did not report this information [9,14,29,30]. Others have included both REP with scaffold and without [11,15,22].

##### Intracanal Coronal Barrier

About intracanal coronal barrier, the level of evidence of the SR’s are of the most part of moderate quality. Despite that, there is no meta-analysis to appraise the usage of intracanal corona barrier on the success of REP treatment.

Mineral trioxide aggregate (MTA) was the most commonly reported. However, there were three studies that did not report the cervical plug [5,13,22], but mentioned the discoloration induced by TAP and MTA [13,22]. Two SRs only contemplate MTA has a coronal barrier, the first included 4 retrospective observational studies without control and one randomized clinical trial (RCT) [7], and the second one case series and three RCTs [27]. On the other hand, one study implicates twenty-nine case reports, seven case series, and two clinical trials that report differentiation between grey and white MTA, and also consider Biodentine and a calcium-enriched mixture [30]. One study compared MTA versus Biodentine as an intracoronal barrier, with MTA causing mild or moderate tooth discoloration compared to Biodentine that had no such disadvantage [29]. One SR that only included RCT reported as a coronal barrier, MTA, glass ionomer cement and resin-modified glass ionomer cement [37]. On the other hand, another study that included three RCTs, six prospective, and two retrospective cohort studies mentioned MTA and Portland cement [36]. Another SR involved 46 studies which included 31 human studies and 15 animal studies. In the first group, an MTA of resin-modified glass ionomer cement, gutta-percha, and calcium-enriched mixture cement was applied as a coronal barrier, instead of the second that used, as an MTA, glass ionomer cement and amalgam [12]. One SR that presented clinical research studies and serial case reports report MTA as a cervical sealing; however, with the goal of facilitating this sealing, some studies modified this technique by applying a matrix of Collaplug or CollaCote and then sealing with MTA [4]. In other studies, a blood clot supplemented with PRP or a blood clot and an injectable scaffold impregnated with basic fibroblast growth factor were used [4]. One study, which included only clinical cases described as cervical barrier MTA, gutta percha, composite resin, or glass ionomer cement [14]. Another SR that included eight randomized controlled studies, five prospective case series, and five retrospective studies referred to grey and white MTA, bioceramic, glass ionomer, or Portland cement as the coronal barrier [15]. Already, in turn, Ong et al. [36] presented eleven studies, three RCTs, six prospective, and two retrospective cohort studies, where MTA or Portland cement were applied [36]. Lastly, Torabinejad et al. [11] included 144 studies of RCT, prospective cohort, retrospective cohort, case series, and case report, where MTA, Biodentine, calcium hydroxide, dycal, cavit + IRM, zinc oxide eugenol, resin modified glass ionomer cement, and glass ionomer cement were indicated [11].

#### 3.4.2. Clinical Outcomes

The level of evidence with respect to clinical outcomes is of low quality, on average (Table 2).

Nineteen SRs report outcomes of REP [4,5,6,7,8,9,12,13,22,23,27,28,29], six being meta-analyses [6,10,11,14,15,36]. Despite all the studies evaluating the effectiveness of pulp revascularization in root formation and the development of necrotic immature permanent teeth, one study approaches if the etiology of pulp necrosis affects the outcome of REP [15]. Another approach observed was the result of endodontic healing with autologous platelet concentration [9,28]; other REPs were performed in only one single visit [29] and were further compared with the outcome of the apexification and REP [5,6,10,11,23]. Several parameters were evaluated such as the etiology of pulp necrosis, periapical pathology resolution, apical closure, increase in root length, and root dentin thickening; the results are promising yet contradictory. Regarding the etiology of pulp necrosis, REP was successful in 94.5% with different etiologies: dental trauma (64.94%), necrosis due to dens evaginatus (22.92%), dental caries (5.61%), and broken cusp with unknown etiology (6.51%) [15]. There was not a significant difference between the results of REP among these teeth with trauma [15]. Regarding apexification versus REP, the results of the clinical success rate are identical. In addition, one SR reports a clinical success rate of 87.9% in the revascularization and 90.6% in the apexification, despite no study performing vitality tests. In the REP, a considerable increase in thickening of the lateral dentinal walls was shown in most of the revascularization cases, while the apical barrier technique with MTA as an apical plug showed an inferior outcome [5]. Likewise, the continuity of dental development was higher in the revascularization when compared to MTA apexification [5]. Other SRs showed no difference between apexification and REP; their results revealed a survival rate of 100% for REP and 95% for apexification [23]. Furthermore, a meta-analysis reported that there was no significant difference between REP with blood clot induction and MTA apexification, and none of the included studies assessed the formation of calcified barrier as an outcome [10]. Two studies stated that in blood clot cases, the most common reasons for failure were reinfection or persistent infection [10]. Another meta-analyses demonstrated survival rates of 97.8% for REP, and 97.1% for apexification, the main treatment complication being crown discoloration [11]. Finally, one SR shows that periapical healing, continued root development, and dentinal thickening of walls diverge according to the intracanal medication, scaffold, and cervical barrier [6]. In some cases, clinical and radiographic examination during the follow-up presented signs and symptoms of failure. Beyond the report of positive outcomes (resolve of periapical radiolucency, increase in root length and root wall thickness, and apical closure), apical closure rates varied across studies [4,6,7,8,12,14,22,27,36]. Regarding intracanal medication, two different medications were compared—TAP and calcium hydroxide associated with CHX—with the results being highly satisfactory. Additionally, taking into consideration apical closure, studies that used distinct techniques of REP showed a reduction in the apical diameter or apical closure in most cases, this reduction being varied according to the technique used, with the best results reached in cases in which induction of bleeding to form an intracanal blood clot was carried out [13]. Two SRs evaluate the efficacy of autologous platelet concentration on endodontic healing, where positive outcomes reported healing of periapical lesions, apical closure, root lengthening, wall thickening, and positive pulp tests [9,28]. Only one SR considered REP in a single visit procedure, which on RCT studies, the clinical outcome was the absence of signs and symptoms, and the radiographic outcome was a decrease in periapical lesion [29]. With regard to animal studies, the histological outcomes were intracanal connective tissue ingrowth in all specimens and beginning of mineralization [29].

#### 3.4.3. Stem Cells Research

The level of evidence with respect to stem cells research was of low quality, on average (Table 2).

Of the twenty-nine SRs analyzed, eight focused on histological assessment and the role of stem cells in REP [24,25,26,31,32,33,34,35], two of them with a meta-analysis performed [34,35]. 

Regarding the role of stem cells, five SRs [25,26,33,34,35] included animal teeth, mostly dog teeth, in their analysis, and only one focused solely on human teeth, specifically, the impacted 3rd molar [31]. 

After the discovery of adult mesenchymal stem cells (MSCs), investigation of its characteristics and potential has spiked. Dental stem cells (DSCs) are MSC-like populations with great multi-differentiation and regeneration potential. Currently, there are five main DSCs: dental pulp stem cells (DPSCs), stem cells from exfoliated deciduous teeth (SHED), stem cells from apical papilla (SCAP), periodontal ligament stem cells (PDLSCs), and dental follicle precursor cells (DFPCs) [39,40,41]. 

Considering cell homing and cell transplantation, four reviews [25,26,34,35] showed inconclusive results and refer that further studies and research are needed, although Eramo et al. [26] mention that cell homing is currently the most clinically available pathway for REP. All four reviews are in agreement that future researchers should apply an accepted and standardized methodology and use a defined set of outcomes that best represent the functional regeneration of pulpal tissues in humans [25,26,34,35]. One review went even further and concluded that functional pulp regeneration can be represented by transplanting stem cells that include vascular and neural regeneration [35].

Two reviews focused on the use of biomaterials and its interactions with the stem cells [31,33]. Both demonstrated that the presence of bioceramic materials can potentially induce mineralization and odontogenic/osteogenic differentiation of human stem cells from the apical papilla, thus prompting their use in REP, which allows for a better incorporation of stem cells and growth factors along with a controlled rate of regeneration [31,33]. Despite these conclusions, they are in agreement that a clear guideline for suitable and preferable biomaterials is in need.

As far as histological outcomes are concerned, one review aimed to evaluate the tissues formed in immature teeth with necrotic and infected pulps after attempted endodontic regeneration procedures [32] and concluded that none of the regeneration protocols resulted in the predictable formation of a true pulp–dentin complex, while another review [24] analyzed the dentine–pulp complex after REP and found that the newly formed tissues indicate tissue repair or a combination of repair and regeneration, with the presence of cementum-like or bone-like instead of dentine, periodontal-like, or pulp tissue.

## 4. Discussion

The present umbrella review aimed to critically gauge the available SRs on REPs, regarding quality of evidence, REP protocols, and clinical outcomes. From the included SRs, the quality of evidence is unfavorable, with only two SRs being of high quality. The majority had critically low, low, or moderate quality of evidence, hence there is an urgent need towards high-quality SRs on REPs.

Our findings may have relevant implications in terms of clinical protocols management and promotion of future investigations to reach precise and scientifically guided clinical protocols. 

Well-conducted SRs on health care interventions that use a predefined, explicit methodology to synthesize the relevant evidence are essential. The inclusion of case reports in SRs is debatable and compromises the integrity of the results (either narrative or meta-analytical estimates). In fact, several SRs have included case reports to accomplish their conclusions [4,8,9,11,12,14,24,26,28,29,30]. The main reason is the fact that REP is an extremely innovative and recent concept, and therefore, there are not so many clinical studies available. The inclusion of clinical cases due to a lack of clinical trials might bias the SR conclusion. On the one hand, through a case report, it is uncertain whether the adverse event or outcome was caused by the intervention. On the other hand, case reports may describe a discrepant intervention, event, or a false alarm that is not as straightforward to evaluate as interventional studies with a calculated number of participants. Therefore, the desire to be “all-inclusive” and the need to avoid publicizing biased or unreliable reports that might trigger a false alarm should be avoided [42]. In the future, authors should first create solid and based research protocols for non-RCTs or RCTs to assess each clinical step of REP protocols.

A key hallmark of a high-quality SRs is the development of a well-described protocol that establishes the main aims, key design characteristics, and planned analyses for the review [42,43]. Although a number of SRs are published unregistered, there are some journals requiring SR protocol registration. Furthermore, SRs conducted under the standards of the Cochrane Collaboration and the 2009 PRISMA statement [19,44] require a priori registration. The protocol registration helps authors to anticipate methodological challenges, to minimize the bias in their conduct and reporting of the review, to reduce duplication, and, above all, provides input on the proposed registration process [19,43]. From this point of view, it is important to highlight that only three included SRs [10,15,37] followed those recommendations. Furthermore, Tong et al. [38] mentioned a registration protocol, but the registration number is not present in the manuscript. Although the absence of registration or of reporting of the register number does not necessarily indicate that the SR was not well conducted, protocol registration must be encouraged [45].

Notwithstanding, some SRs included, simultaneously, studies in humans and studies in animal models [12,22,26,34]. Without a doubt, animal research is essential to biological research and has historically been employed towards drug approval processes [46]. However, due to the uncertainty of whether animal studies are reliable scientific sources for human situations, there is some reluctance as to the ethics of this rationale [47]. For this reason, the acceptability of SRs that mix clinical (say, in humans) and pre-clinical (say, in animal models) is highly debatable. None of these SRs combined data from preclinical and clinical data, and two SRs [26,34] have focused on dental pulp cells in particular. Yet, the confidence in such studies is always undermined for the reasons aforementioned.

As far as risk of bias is concerned, some SRs showed an inconsistency when applying instruments to assess the methodological quality of the included studies [7,12,13,27,28,36,38]. For instance, the use of instruments to assess randomized trials in case reports or non-randomized trials occurred and might certainly lead to spurious conclusions. Moreover, the consequence of this lapse was the downsizing of studies’ risk of bias, for example, assessing randomization and blinding properties in non-randomized studies where such steps do not exist.

Regarding meta-analytical estimates, the majority of studies employed approaches that demand cautiousness. While meta-analysis is the statistical combination of two or more studies, the results might be spurious with too narrow confidence intervals when there are not enough studies (normally at least five studies included) and this demands carefulness when interpreting results.

From the clinical point of view, the management of immature permanent teeth with necrotic pulp is a challenging clinical procedure. Traditionally, apexification was the first choice for dealing with these situations. However, REP grounds on three main objectives: (1) apexification, to prevent or to heal the periapical tissues; (2) the increase of length and thickness of the root, increasing the root resistance to fracture; and (3) to regain pulp sensitivity [24,25,26,31,32,33,34,35]. Overall, the available evidence points to the existence of clinical success in these three premises; however, the scientific robustness regarding the most appropriate clinical protocol remains to be explained. On the other hand, the so-called REP is, in fact, a repairing/healing procedure rather than a regenerative one, considering the tissues and cell populations that derive from it [48,49]. Hence, tissue and stem cell engineering will be central to achieve regeneration per se.

### Strengths and Limitations

The present umbrella review has several strengths. Overall, these results provide a comprehensive overview of the available SRs on REP using a transparent and evidence-based methodology. We commend a cautious interpretation, as the individual studies included in each of the present SR were not explored. Thus, the conclusions lean on the interpretation of the systematic review’s authors. Another point worth mentioning is the existence of two PROSPERO registers that are in an ongoing status, but not published. 

## 5. Conclusions

The quality of evidence produced by the available SRs was not favorable. Future high standard SRs and well-designed clinical trials are warranted to better clarify the clinical protocols and outcomes of success of REP.

## Figures and Tables

**Figure 1 ijerph-18-00754-f001:**
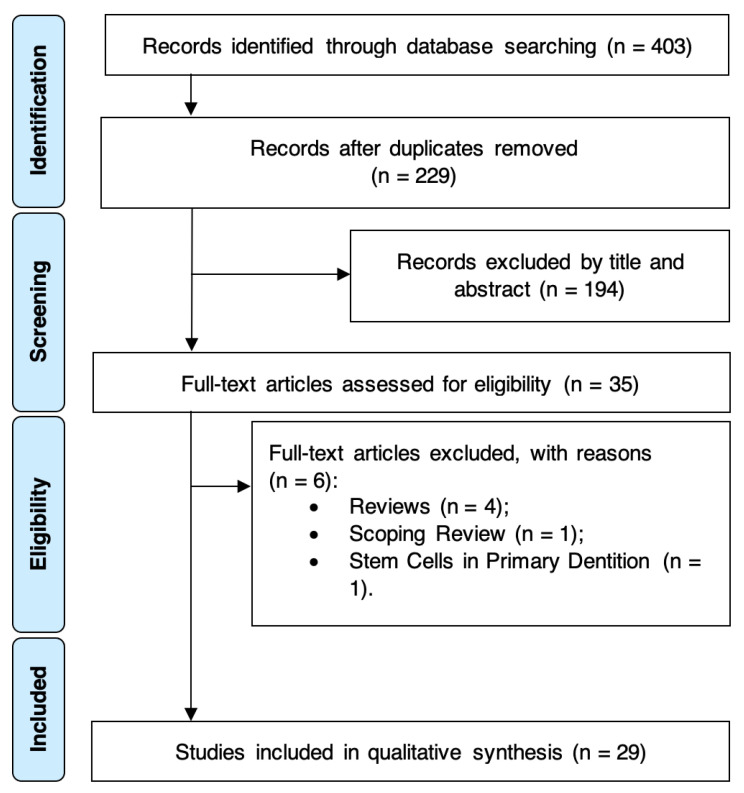
PRISMA diagram showing the exclusion and inclusion process of the literature review.

**Table 1 ijerph-18-00754-t001:** Baseline characteristics.

Author (Year)	Risk of Bias	Search Period	Interventions	Types/No. of Studies Included	Number of Teeth	Tool Used for Quality Assessment	Method of Analysis	Outcomes	Findings
Antunes 2015 [4]	Critically low	Up to July 2014	Effectiveness of REP in root formation of necrotic immature permanent teeth	11 studies (clinical research studies and case reports)	Human = 176; Animal = 0	NR	SR	REP techniques stimulate the development of the apical closure and thickening of radicular dentin. Several aspects still remain unknown.	The evidence should be interpreted with caution as the articles report different methods and evaluation parameters.
El-Sayed 2015 [25]	Moderate	November 1971 until July 2014	Stem cell transplantation for REP	2 RCTs 3 non-RCTs	Human = 0; Animal = 222	SYRCLE Guidelines	SR	The results should be interpreted with caution. Future studies should apply an accepted and standardized methodology that best represent functional regeneration of pulpal tissues.	Stem/progenitor cell transplantation seems to enhance pulp–dentin complex regeneration in intraoral animal models in vivo.
He 2017 [14]	Critically low	Until September 2016	Treatment of necrotic teeth by apical revascularization	36 case reports	Human = 36; Animal = 0	NR	SR/MA	Lacks consistency in promoting root lengthening, widening or apical closure.	Apical revascularization facilitates tooth-root development.
Bucchi 2017 [22]	Critically low	Up to May 9, 2016	Clinical protocols used for REP of immature necrotic teeth	11 clinical human studies 6 case reports 5 pilot clinical studies 12 animal studies	Human = 222; Animal = 275	NR	SR	Due to the heterogeneity of the analyzed studies, it was not possible to quantitatively analyze the influence of agents, their concentrations and time for application on the clinical, radiographic and histological outcomes.	It is necessary to conduct clinical and animal studies to establish if the protocol described is related to better clinical, histological and radiographic outcomes.
Cabral 2016 [23]	Critically low	January 2000 until June 2015	Treatment of immature teeth with apical periodontitis after REP	2 RCTs	Human = 92; Animal = 0	NR	SR	REP allow a greater possibility of continuity of root formation than traditional apexification procedures.	The scientific evidence should be interpreted with caution since there were only two studies included.
Tong 2017 [38]	Critically low	Up to March 25, 2016	REP in the management of non-vital immature permanent teeth	14 clinical studies	Human = 389; Animal = 0	NOS—cohort and case–control studies. The Cochrane RoB tool – RCYs and non-RCTS	SR/MA	Many knowledge gaps still exist within the studies published.	Excellent success rates in terms of tooth survival and resolution of periapical pathology after REP. There were inconsistent results for more desirable outcomes.
Koc 2020 [15]	High	January 2014 to June 2019	Which tooth is treated with REP	8 RCTs 5 case series 5 retrospective studies	Human = 445; Animal = 0	Modified Cochrane Collaboration tool	SR/MA	The results should be evaluated with caution because information about the irrigation time for each solution used during the treatment, the presence of periapical lesion, and how long the tooth had been infected is lacking.	There is no evidence to support the hypothesis that the etiology of pulp necrosis may affect the outcome of REP.
Eramo 2017 [26]	Critically low	Up to 2016	Cell homing for REP	10 studies	Human = NR; Animal = NR	NR	SR	Cell homing currently represents the most clinically viable pathway for dental pulp regeneration.	Cell homing strategies for pulp regeneration need further understanding and improvement if they are to become a reliable and effective approach in endodontics.
Duggal 2017 [6]	Low	Since 1966 up to 2017	Management of non-vital permanent anterior teeth with incomplete root development	6 studies	Human = 538; Animal = 0	Cochrane RoB tool	SR	REP is currently extremely weak and this technique should only be used in very limited situations.	The current review supports the use of MTA followed by root canal obturation as the treatment of choice.
Santos 2018 [30]	Critically low	Up to March 2017	Alternative materials to conventional TAP and grey MTA could avoid tooth discoloration in teeth submitted to REP	29 case reports 7 case series 2 RCTs	Human = 189; Animal = 0	NR	SR	The sole effect of the different materials involved in REP on tooth discoloration is a very hard task, since intracanal medication and cervical sealing are applied sequentially, and both have potential to induce tooth color alteration.	The use of alternative materials to TAP and grey MTA reduces the occurrence of tooth discoloration.
Alghamdi 2020 [12]	Low	2009-2019	Clinical protocols of REP in the management of immature permanent teeth with necrotic pulp	31 human studies 15 animal studies(RCT, case reports, in vitro with in vivostudies, in vivo studies, prospective and retrospective studies)	Human = 469; Animal = 537	Cochrane RoB tool	SR	REP showed better results in certain parameters in the management of immature necrotic permanent teeth.	More clinical trials with a standardized protocol and defined clinical, radiographic, and histopathological outcomes with longer follow-up periods are warranted.
Altaii 2017 [32]	Critically low	Up to mid-July 2016	Histological tissues assessment in immature animal teeth with necrotic and infected pulps after REP using different scaffolds	13 studies	Human = 0; Animal = 309	NR	SR	None of the REP resulted in the predictable formation of a true pulp–dentin complex.	The formation of highly organized and functional pulp and dentin remains a challenging problem in immature teeth with necrotic and infected pulps.
Panda 2020 [37]	High	From 2012 until 2020	Effectiveness of autologous platelet concentrates compared to blood-clot regeneration in non-vital immature permanent teeth	10 RCTs	Human = 321; Animal =0	Selection bias, performance bias, detection bias, attrition bias, and reporting bias	SR/MA	Autologous platelet concentrates could be beneficial to improve apical closure and response to vitality tests.	Further studies with standardized protocols are necessary to assess the actual contribution of autologous platelet concentrates in REP.
Kontakiotis 2014 [8]	Low	January 1993 to the 2nd week of December 2013	REP	2 cohort studies 8 case series 41 case reports	Human = 255; Animal = 0	NOS	SR	The current best available evidence allows clinicians to provide this treatment modality safely to patients.	REP is considered to be a safe and effective treatment option.
Meschi 2016 [28]	Moderate	12 June 2015 and updated on 16 January 2016	The impact of autologous platelet concentrates on endodontic healing	7 RCTs 41 non-RCTs	Human = 279; Animal = 0	Cochrane Collaboration tool	SR	There is a huge lack of standardization in treatment protocols and long-term high-quality clinical trials.	Autologous platelet might accelerate postoperative bone healing, improve the patients’ QoL in the early postoperative period, aid further root development, and support maintenance or regaining of pulp vitality.
Bakhtiar 2018 [33]	Critically low	Since 2010	Stem cell therapy to regenerate the dentine–pulp complex and the success of clinical protocols	53 studies	Human teeth = NR; Animal teeth = NR	NR	SR	Scaffolds and biomaterials provide a meaningful approach to better incorporate stem cells and growth factors along with controlled rate of regeneration.	Future studies are needed to focus on providing a clear guideline for suitable and preferable properties of biomaterials to be used in REP.
Lolato 2016 [27]	Moderate	From 2000 up to November 2015	Platelet concentrates for revitalization of immature necrotic teeth	1 case series 3 RCTs	Human = 61; Animal =0	NR	SR	Platelet concentrates showed promising results that warrant further investigation.	Autologous platelet concentrate has potential in promoting root development of necrotic immature teeth.
Digka 2019 [24]	Critically low	Up to January 2019	Regeneration of the dentine–pulp complex through the neo-deposition of dental and pulpal tissues	12 studies	Human = 14; Animal =0	NR	SR	In immature permanent human teeth treated with REP, the newly formed tissues indicate tissue repair or a combination of repair and regeneration.	Further clinical and histological research is necessary in order to establish an appropriate treatment protocol related to the pretreatment status of the dental pulp and the periapical tissues.
El-Sayed 2019 [34]	Moderate	Up to January 2019	Effect of stem/progenitor cells’ transplantation on pulpal tissue regeneration, apical healing and pulpal vitality	8 animal studies (2 RCTs, 7 non-RCTs) 1 human RCT	Human = 40; Animal =336	Cochrane RoB tool	SR/MA	The transplantation of stem/progenitor cells shows promise for pulp regeneration whilst clinical routine application appears to be currently still not in reach.	Significant methodological heterogeneity was identified across studies.
Torabinejad 2017 [11]	Moderate	From June 1966 until November 2016	Clinical outcomes of REP and MTA apical plug	144 studies	Human = 998; Animal =0	Cochrane Collaboration’s tool	SR/MA	The existing literature lacks high-level clinical studies. More studies with large sample sizes and long-term follow-ups are needed	The treatment of immature teeth with pulp necrosis using an MTA apical plug or REP results in high survival and success rates.
Nicoloso 2019 [10]	Moderate	From 2012 to 2017	REP for the Treatment of Immature Necrotic Permanent Teeth	3 retrospective cohort studies	Human = 135; Animal =0	NOS	SR/MA	The results do not favor one treatment modality over the other. More clinical studies are necessary.	The current literature regarding the clinical, radiographic and functional retention outcomes in immature necrotic permanent teeth treated either with pulp REP or apexification is limited.
Chisini 2018 [5]	Moderate	Up to July 30, 2017	Performing revascularization relying on blood clot formation after induced periapical bleeding	3 retrospective studies, 2 prospective studies 1 RCT	Human = 155; Animal =0	Cochrane RoB tool	SR	Clinical success of therapies, deposition and thickening of lateral dentinal walls and the continuation of root development.	The results should be interpreted with caution, despite the apexification with MTA-apical plug provides similar clinical success to REP, since the radiographic measurements showed an improvement in thickening of lateral dentinal walls.
Jamali 2020 [35]	Moderate	Between 2010 and 2019	Stem cell-mediated REP	5 animal studies 1 human study	Human = 26; Animal =194	Cochrane collaboration tool	SR/MA	The use of dental stem cells in regenerating and repairing teeth as well as their differentiation potentials.	Promising parameters testing functional pulp regeneration can be represented by transplanting stem cells that include vascular and neural regeneration.
Couto 2019 [13]	Moderate	Up to February 2017	REP with TAP	1 RTC 7 RCTs with control group	Human = 159; Animal =0	Cochrane RoB tool	SR	TAP is effective in the pulp REP of teeth with incomplete root formation.	It was demonstrated that a scarcity of studies performed pulp revascularization procedures using TAP as an intracanal medication.
Sanz 2020 [31]	Moderate	Up to December 2019	Viability and stimulation of human stem cells from the apical papilla	10 studies	Human = NR; Animal =NR	Consort Checklist	SR	Both bioceramic materials showed significant positive results when compared to a control for hSCAP cell viability, migration, and proliferation assays.	Commercially available silicate-based materials can potentially induce mineralization and odontogenic/osteogenic differentiation of human stem cells from the apical papilla.
Metlerska 2019 [9]	Moderate	NR	Efficacy of autologous platelet concentrates in REP	5 RCTs 21 case reports	Human = 37; Animal =0	Cochrane Collaboration’s tool	SR	Autologous platelet concentrates can lead to development of the root and protect the tooth from extraction. However, more long-term clinical studies are needed.	Procedures using autologous platelet concentrates contribute to the success of treating immature permanent teeth.
Rossi-Fedele 2019 [29]	Moderate	From their inception to July 2018	Benefits of single visit of REP	5 case reports 1 RCT 1 animal study	Human = NR; Animal =28	Cochrane RoB tool—RCTs SYRClES tool—animal studies	SR	Successful single-visit REP commonly includes the use of high concentrations of sodium hypochlorite and EDTA combined with the use of agitation systems.	The evidence supporting the potential use of single-visit REP is scarce.
Ong 2020 [36]	Moderate	Since 1990 until 2019	Appraise the level of evidence of the existing in REP	3 RCTs 6 prospective cohort studies 2 retrospective cohort studies	Human = 282; Animal =0	NOS—observational studies Cochrane RoB tool—RCTs and non-RCTs	SR / MA	REP yielded high survival and healing rates with a good root development rate.	Clinical meaningful root development after REP remained unpredictable.
Kharchi 2020 [7]	Moderate	1 January 2004 until 24 April 2020	Clinical and radiographic outcomes of REP involving any disinfection irrigant or antibiotic	4 Retrospective observational without control 1 RCT	Human = 70; Animal =0	Cochrane RoB tool— RCTs Quality Assessment Tool for Quantitative Studies—Observational studies	SR	REP using a non-antibiotic disinfectant approach appears capable of providing satisfactory outcomes for a non-vital immature permanent tooth.	REP is an advancing area of dentistry with great potential, but more long- term, robust and high levels of evidence are required to provide further recommendations.

EDTA—Ethylenediaminetetraacetic acid; hSCAP—human Stem Cells from the Apical Papilla; MA—Meta-Analysis; MTA—Mineral Trioxide Aggregate; NOS—Newcastle—Ottawa scale; NR—Not Reported; QoL—Quality of Life; RCT—Randomized Clinical Trials; REP—Regenerative Endodontic Procedure; RoB—Risk of Bias; SR—Systematic Review; SYRCLE—Systematic Review Center for Laboratory animal Experimentation; TAP—Triple Antibiotic Paste.

**Table 2 ijerph-18-00754-t002:** Risk of Bias of Systematic Reviews (AMSTAR 2 tool).

Author (Year)	1	2	3	4	5	6	7	8	9	10	11	12	13	14	15	16	Review Quality
Kontakiotis 2014 [7]	Y	Y	N	PY	N	N	PY	PY	N/PY	N	0/0	0	Y	N	0	Y	Low
Antunes 2015 [4]	Y	PY	Y	PY	Y	Y	N	PY	N/N	N	0/0	0	N	N	0	Y	Critically Low
El Sayed 2015 [25]	Y	PY	N	Y	Y	Y	PY	N	PY/0	N	0/0	0	Y	Y	0	Y	Moderate
Cabral 2016 [23]	Y	N	N	PY	Y	Y	PY	N	0/N	N	0/0	0	N	N	0	Y	Critically Low
Lolato 2016 [27]	Y	PY	Y	PY	Y	Y	PY	N	PY/0	N	0/0	0	Y	N	0	Y	Moderate
Meschi 2016 [28]	Y	PY	Y	PY	Y	Y	Y	PY	PY/PY	N	0/0	0	Y	Y	0	Y	Moderate
Altaii 2017 [32]	Y	PY	Y	PY	Y	N	N	Y	N/N	N	0/0	0	N	N	0	Y	Critically Low
Bucchi 2017 [22]	Y	N	N	PY	Y	Y	N	N	N/0	N	0/0	0	N	Y	0	Y	Critically Low
Duggal 2017 [6]	Y	PY	Y	PY	Y	Y	PY	N	N/0	N	0/0	0	N	Y	0	Y	Low
Eramo 2017 [26]	Y	PY	N	PY	Y	Y	PY	N	N/N	N	0/0	0	N	N	0	Y	Critically Low
He 2017 [14]	Y	PY	Y	PY	Y	Y	Y	N	0/N	N	0/N	0	N	N	0	Y	Critically Low
Tong 2017 [38]	Y	PY	Y	N	Y	Y	PY	N	PY/N	N	Y/0	Y	Y	Y	Y	Y	Critically Low
Torabinejad 2017 [11]	Y	Y	Y	PY	Y	Y	Y	PY	PY/0	N	Y/0	Y	Y	Y	N	Y	Moderate
Bakhtiar 2018 [33]	Y	N	N	N	N	N	N	Y	0/N	N	0/0	0	N	N	0	Y	Critically Low
Couto 2019 [13]	Y	PY	Y	PY	Y	Y	Y	PY	Y/Y	N	0/0	0	Y	Y	0	N	Moderate
Chisini 2018 [5]	Y	PY	Y	PY	Y	Y	PY	PY	PY/PY	N	0/0	0	Y	Y	0	Y	Moderate
Santos 2018 [30]	Y	Y	Y	PY	Y	Y	PY	N	0/N	N	0/0	0	N	N	0	N	Critically Low
Digka 2019 [24]	Y	PY	Y	PY	Y	Y	PY	PY	N/N	N	0/0	0	N	N	0	Y	Critically Low
El Sayed 2019 [34]	Y	PY	Y	PY	Y	Y	PY	PY	PY/PY	N	Y/Y	Y	Y	Y	N	Y	Moderate
Metlerska 2019 [9]	Y	PY	N	PY	Y	Y	Y	PY	PY/0	N	0/0	0	Y	Y	0	Y	Moderate
Nicoloso 2019 [10]	Y	PY	Y	PY	Y	Y	PY	PY	0/PY	N	0/Y	Y	Y	Y	Y	N	Moderate
Rossi-Fedele 2019 [29]	Y	PY	Y	PY	Y	Y	N	PY	PY/PY	N	0/0	0	Y	N	0	N	Moderate
Alghamdi 2020 [12]	Y	PY	N	PY	Y	Y	N	N	PY/PY	N	0/0	0	N	N	0	Y	Low
Jamali 2020 [35]	N	PY	N	PY	N	N	N	Y	PY/Y	N	Y/Y	Y	Y	Y	Y	Y	Moderate
Kharchi 2020 [7]	Y	PY	N	PY	N	N	Y	Y	PY/PY	N	0/0	0	Y	Y	0	N	Moderate
Koc 2020 [15]	Y	Y	Y	PY	Y	Y	PY	Y	PY/0	Y	Y/0	Y	Y	Y	Y	Y	High
Ong 2020 [36]	Y	Y	N	PY	Y	Y	N	PY	PY/PY	N	Y/Y	Y	Y	Y	Y	Y	Moderate
Panda 2020 [37]	Y	PY	Y	PY	Y	Y	Y	PY	PY/0	N	Y/0	Y	Y	Y	Y	Y	High
Sanz 2020 [31]	Y	PY	Y	PY	Y	Y	PY	PY	PY/PY	N	0/0	0	Y	N	0	Y	Moderate

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
