# Peer review of "Regenerative Endodontic Procedures: An Umbrella Review"

_ijerph, 2021, doi:10.3390/ijerph18020754_

Round 1
Reviewer 1 Report
This manuscript reviews the published SRs on REP. The following revisions are needed:
- Line 50-53: This seems not relevant to the content of the manuscript. Please remove it.
- Rephrase Line 201-204, 248-250, 256-259, 260-263, 379-383
- Cite the sentence from Line 211-214, 244-246, 250-252, 272-274, 361-362, 374-377
- Rewrite and cite the sentence from Line 238-340, and define what is the meaning of broken central cusp.
- There are discrepancies between the objective of the review, and the result/conclusion of this review.
- There are some punctuation, abbreviation, capitalization and grammatical errors found throughout the manuscript.
- Generally, the authors make a good effort in reviewing the SRs on REP. However, some of the contents in both Results and Discussions are not clearly written or cited. Thus, a major revision is required.
- Professional proofread is advised.
Author Response
To the Editorial Board of
International Journal of Environmental Research and Public Health
We are pleased with the opportunity to revise and resubmit our manuscript titled “Regenerative Endodontics Procedure: An Umbrella Review” (Manuscript ID ijerph-1069768).
We have considered all editorial and reviewers’ comments and incorporated changes in the new revised version of the manuscript. Please find enclosed a track-changes draft of the manuscript and in addition a point-by-point rebuttal to all comments raised as outlined below. We hope that you find our responses satisfactory in addressing the criticisms and suggestions.
We hope the revised manuscript will be in acceptable format for your journal.
Referee: 1
This manuscript reviews the published SRs on REP. The following revisions are needed:
Line 50-53: This seems not relevant to the content of the manuscript. Please remove it.
Answer: We appreciate and apologize for this typo. We have removed it from the manuscript.
Rephrase Line 201-204, 248-250, 256-259, 260-263, 379-383
Answer: Regarding these particular lines:
Lines 201-204 now reads: “One study described single-visits REP, comparing MTA and Biodentine versus MTA and silver amalgam [29].”
Lines 248-250 now reads: “Another systematic review showed no differences between Apexification and REP or between 95% maintenance of revascularized teeth and 100% with apexification [25].”
Lines 256-259 now reads: “Finally, one systematic review showed that periapical healing, continued root develop-ment and dentinal thickening of walls diverge according to the intracanal medication, scaffold and cervical barrier”.
Lines 260-263 now reads: “In some cases, clinical and radiographic examination during follow-up presented signs and symptoms of failure. Beyond the report of positive outcomes (resolve of periapical radiolucency, increase in root length and root wall thickness, and apical closure), apical closure rates varied across studies [4,6–8,12,14,22,27,36].”
Lines 379-383 now reads:”On the other hand, the so-called REP is, in fact, a repairing/healing procedure rather than a regenerative one, considering the tissues and cell populations that derive from it [48,49]. Hence, tissue and stem cell engineering will be central to achieve regeneration per se [48,49].”
Cite the sentence from Line 211-214, 244-246, 250-252, 272-274, 361-362, 374-377
Answer: We are thankful for this important remark. We have cited the referred sets of lines accordingly:
Line 211-214 - One SR that presented clinical research studies and serial case reports report MTA as a cervical sealing, however with the goal of facilitating this sealing, some studies modified this technique by applying a matrix of Collaplug or CollaCote and then sealing with MTA [4].
Line 244-246 - In the REP a considerable increase in thickening of lateral dentinal walls was shown in most of revascularization cases, while apical barrier technique with MTA as apical plug showed an inferior outcome [5].
Line 250-252 - Furthermore, a meta-analysis reported that there was no significant difference between REP with blood clot induction and MTA apexification, and none of the included studies assessed the formation of calcified barrier as an outcome [10].
Line 272-274 - Only one systematic review consider REP in a single visit procedure, which on RCT studies the clinical outcome was absence of signs and symptoms, and the radiographic outcome was a decrease of periapical lesion [29].
Line 361-362 - What risk of bias concerns, some SRs showed an inconsistency when applying instruments to assess the methodological quality of the included studies [7,12,13,27,28,36,38].
Line 374-377 - Albeit REP grounds on three main objectives: 1) apexification, to prevent or to heal the periapical tissues; 2) the increase of length and thickness of the root, increasing the root resistance to fracture; and, 3) to regain pulp sensitivity [24-26,31-35].
Rewrite and cite the sentence from Line 238-340, and define what is the meaning of broken central cusp.
Answer: We followed your suggestion to rewrite, cite and define “broken central cusp”: “Regarding the etiology of pulp necrosis, REP was successful in 94.5% with multiple etiologies: dental trauma (64.94%); necrosis due to dens evaginatus (22.92%); dental caries (5.61%); and broken cusp with unknown etiology (6.51%) [15].”
There are discrepancies between the objective of the review, and the result/conclusion of this review.
Answer: We fully agree with the reviewer that there is a discrepancy between the review objectives and the results/conclusions. In fact, two objectives are present in the introduction, and the last one is not correct. We have removed the misleading paragraph on the objectives and, therefore, we consider that the conclusions respond to the objectives of the review.
There are some punctuation, abbreviation, capitalization and grammatical errors found throughout the manuscript.
Answer: We have read carefully the entire manuscript to resolve all punctuation, abbreviation, capitalization and grammatical errors found throughout the manuscript.
Generally, the authors make a good effort in reviewing the SRs on REP. However, some of the contents in both Results and Discussions are not clearly written or cited. Thus, a major revision is required.
Answer: Taking in consideration all your commentaries, we hope have left Results and Discussion better clearly written and cited.
Professional proofread is advised.
Answer: We have followed your suggestion.
Reviewer 2 Report
Thank you for the interesting article.
My main concern is that the Materials and methods AND results section are okay, but the introduction and the discussion need to be adapted better to each other and the M&M and results.
The part of the aim about the clinical outcomes is not answered in the conclusions.
In the aim: With the present study, we intend to evaluate a relationship between measurements of BP and tooth loss and the age mediation effect, on a large cross-sectional study from a national reference dental care clinic. As a second aim, we investigated the prevalence of potential undiagnosed hypertension patients.
This is not discussed in the article and especially not in the conclusion. Can it be removed? Or is something missing in the article?
Some other remarks, more related to the text of the article.
Introduction
“biologically based procedures designed to replace damage structure” -> damaged
Therefore, REP have the potential to increase root length, thickening of the root wall and apical closure, and therefore be an alternative approach for necrotic immature permanent teeth [4–13]. -> to increase root length, to thicken the rootwall and to achieve apical closure.
As a second aim, we investigated the prevalence of 53 potential undiagnosed hypertension patients. -> this second aim seems to come out of the blue for me. Please add some more backgroundinformation in about hypertension and REP.
There is nothing about hypertension in the results and discussion. So remove the second aim.
You describe the differences in treatment protocols, but do not say what is most often used/advised. Why?
Why are both clinical and animal studies included? I think they are difficult to compare.
Author Response
To the Editorial Board of
International Journal of Environmental Research and Public Health
We are pleased with the opportunity to revise and resubmit our manuscript titled “Regenerative Endodontics Procedure: An Umbrella Review” (Manuscript ID ijerph-1069768).
We have considered all editorial and reviewers’ comments and incorporated changes in the new revised version of the manuscript. Please find enclosed a track-changes draft of the manuscript and in addition a point-by-point rebuttal to all comments raised as outlined below. We hope that you find our responses satisfactory in addressing the criticisms and suggestions.
We hope the revised manuscript will be in acceptable format for your journal.
Referee: 2
Thank you for the interesting article.
My main concern is that the Materials and methods AND results section are okay, but the introduction and the discussion need to be adapted better to each other and the M&M and results.
Answer: We are very pleased with the interest shown in the article, and we appreciate your comments, all of which are important and pertinent.
The part of the aim about the clinical outcomes is not answered in the conclusions.
Answer: We fully agree with the reviewer that there is a discrepancy between the review objectives and the conclusions. In fact, two objectives are present in the introduction, and the last one is not correct. We have removed the misleading paragraph on the objectives and, therefore, we consider that the conclusions respond to the objectives of the review.
In the aim: With the present study, we intend to evaluate a relationship between measurements of BP and tooth loss and the age mediation effect, on a large cross-sectional study from a national reference dental care clinic. As a second aim, we investigated the prevalence of potential undiagnosed hypertension patients.
This is not discussed in the article and especially not in the conclusion. Can it be removed? Or is something missing in the article?
Answer: We appreciate and apologize for this typo. We have removed it from the manuscript.
Some other remarks, more related to the text of the article.
Introduction
“biologically based procedures designed to replace damage structure” -> damaged
Answer: We followed your suggestion and rewrite
Therefore, REP have the potential to increase root length, thickening of the root wall and apical closure, and therefore be an alternative approach for necrotic immature permanent teeth [4–13]. -> to increase root length, to thicken the rootwall and to achieve apical closure.
Answer: We followed your suggestion and rewrite
As a second aim, we investigated the prevalence of 53 potential undiagnosed hypertension patients. -> this second aim seems to come out of the blue for me. Please add some more background information in about hypertension and REP.
Answer: We appreciate and agree with this part, We apologize for this typo
There is nothing about hypertension in the results and discussion. So remove the second aim.
Answer: We appreciate and agree this part, We apologize for this typo
You describe the differences in treatment protocols, but do not say what is most often used/advised. Why?
Answer: we appreciate this important part. We followed your suggestion, so we rephrased to “Nevertheless, there is no unanimity on the disinfection/irrigation, intracanal medication and scaffolds. For this reason, ESE [16] and AAE [17,18] have recently delivered position statements and clinical considerations based on the best yet limited available evidence. “
Why are both clinical and animal studies included? I think they are difficult to compare.
Answer: We appreciate this part, and we totally agree with the reviewer. Nevertheless, some systematic reviews included clinical and animal studies and it is important to highlight this fact.
Round 2
Reviewer 1 Report
The authors have modified most of the issues listed before. However, there is still slight corrections/clarification needed:
Line 137-152: The punctuations used are inconsistent. For eg, some figures used decimal points (0.2%), some used decimal commas (0,12%).
Line 200-201: One study described single-visits 201 REP, comparing MTA and Biodentine versus grey MTA and silver amalgam.
*What are they comparing? the outcome?
Line 247-248: Did you mean the maintenance/survival rate when you mentioned 95%/100%? If so, please mention that. Also, wrong citation. El-Sayed et al. (reference 25) did not compare the REP and Apexification. Please cite the correct article.
*Please check the references and make sure the sequence of citations did not change after your revision.
Author Response
Esteemed Referee 1, below please find our answers to your commentaries.
Referee: 1
The authors have modified most of the issues listed before. However, there is still slight corrections/clarification needed:
Line 137-152: The punctuations used are inconsistent. For eg, some figures used decimal points (0.2%), some used decimal commas (0,12%).
Answer: We appreciate this remark. We have rewritten to:
“In all, SRs reported several irrigating solutions at distinct concentrations and during different periods of time except two SRs [13,30]. Rossi Fedele et al. [29] suggested that sodium hypochlorite (NaOCl) at different concentrations is the irrigating solution most used in these procedures. Also, in the successful single-visit reported cases, the irrigation included both 2.5% NAOCL, saline and 17.0% ethylenediaminetetraacetic acid (EDTA) [29]. Nevertheless, irrigation with EDTA was not commonly used in studies including animal teeth [22,29]. Concerning two-visit REP, NaOCL at 2.5% is the most employed despite the disinfection using a percentage of 5.25% to 6.0% of NaOCL associated with 0.2 to 2.0% of chlorhexidine (CHX) were also established in non-vital immature permanent tooth [4,7,14]. Other irrigation methods used NaOCL in concentrations that varied from 0.5 to 8.0% isolated or included with sterile saline, 5.0 to 17.0% of EDTA and 0.12 to 2.0% of CHX [5,9,12,15,36,37]. Furthermore, Torabinejad et al. [11] and Lobato el al. [27] reported studies with irrigation protocols with 1.0 to 5.25% of NaOCl isolated or in combination with a 0.12% CHX, or 0.9% saline or 2.0% CHX isolated are also used in REP [11,27]. Additionally, Lobato el al. [27] record respecting irrigating solution is NaOCL at 2.5% solo or in combination with 0.12% CHX and 5.0% EDTA.”
Line 200-201: One study described single-visits 201 REP, comparing MTA and Biodentine versus grey MTA and silver amalgam.
*What are they comparing? the outcome?
Answer: We have rephrased to “One study compared MTA Biodentine as an intra-coronal barrier, with MTA causing mild or moderate tooth discoloration compared to Biodentine that had no such disadvantage [31].”
Line 247-248: Did you mean the maintenance/survival rate when you mentioned 95%/100%? If so, please mention that. Also, wrong citation. El-Sayed et al. (reference 25) did not compare the REP and Apexification. Please cite the correct article.
Answer: Thank you for your remark. We have rephrased to “Other SR showed no difference between Apexification and REP; their results revealed a survival rate of 100% for REP and 95% for Apexification [23].”
*Please check the references and make sure the sequence of citations did not change after your revision.
Answer: We have confirmed the sequence of citations after the revision of all points.
Reviewer 2 Report
thank you for the improvements you made
Author Response
Thank you for your time and effort in reviewing our manuscript. We believe you have certainly strengthened it with you important remarks.